# A Unified Approach to Interpreting Model Predictions

**Scott M. Lundberg**
Paul G. Allen School of Computer Science
University of Washington
Seattle, WA 98105
slund1@cs.washington.edu

**Su-In Lee**
Paul G. Allen School of Computer Science
Department of Genome Sciences
University of Washington
Seattle, WA 98105
suinlee@cs.washington.edu

## Abstract

Understanding why a model makes a certain prediction can be as crucial as the prediction's accuracy in many applications. However, the highest accuracy for large modern datasets is often achieved by complex models that even experts struggle to interpret, such as ensemble or deep learning models, creating a tension between *accuracy* and *interpretability*. In response, various methods have recently been proposed to help users interpret the predictions of complex models, but it is often unclear how these methods are related and when one method is preferable over another. To address this problem, we present a unified framework for interpreting predictions, SHAP (SHapley Additive exPlanations). SHAP assigns each feature an importance value for a particular prediction. Its novel components include: (1) the identification of a new class of additive feature importance measures, and (2) theoretical results showing there is a unique solution in this class with a set of desirable properties. The new class unifies six existing methods, notable because several recent methods in the class lack the proposed desirable properties. Based on insights from this unification, we present new methods that show improved computational performance and/or better consistency with human intuition than previous approaches.

## 1 Introduction

The ability to correctly interpret a prediction model's output is extremely important. It engenders appropriate user trust, provides insight into how a model may be improved, and supports understanding of the process being modeled. In some applications, simple models (e.g., linear models) are often preferred for their ease of interpretation, even if they may be less accurate than complex ones. However, the growing availability of big data has increased the benefits of using complex models, so bringing to the forefront the trade-off between accuracy and interpretability of a model's output. A wide variety of different methods have been recently proposed to address this issue [5, 8, 9, 3, 4, 1]. But an understanding of how these methods relate and when one method is preferable to another is still lacking.

Here, we present a novel unified approach to interpreting model predictions.[1] Our approach leads to three potentially surprising results that bring clarity to the growing space of methods:

1. We introduce the perspective of viewing *any* explanation of a model's prediction as a model itself, which we term the *explanation model*. This lets us define the class of *additive feature attribution methods* (Section 2), which unifies six current methods.

2. We then show that game theory results guaranteeing a unique solution apply to the *entire class* of additive feature attribution methods (Section 3) and propose *SHAP values* as a unified measure of feature importance that various methods approximate (Section 4).

3. We propose new SHAP value estimation methods and demonstrate that they are better aligned with human intuition as measured by user studies and more effectually discriminate among model output classes than several existing methods (Section 5).

## 2    Additive Feature Attribution Methods

The best explanation of a simple model is the model itself; it perfectly represents itself and is easy to understand. For complex models, such as ensemble methods or deep networks, we cannot use the original model as its own best explanation because it is not easy to understand. Instead, we must use a simpler *explanation model*, which we define as any interpretable approximation of the original model. We show below that six current explanation methods from the literature all use the same explanation model. This previously unappreciated unity has interesting implications, which we describe in later sections.

Let $f$ be the original prediction model to be explained and $g$ the explanation model. Here, we focus on *local methods* designed to explain a prediction $f(x)$ based on a single input $x$, as proposed in LIME [5]. Explanation models often use *simplified inputs* $x'$ that map to the original inputs through a mapping function $x = h_x(x')$. Local methods try to ensure $g(z') \approx f(h_x(z'))$ whenever $z' \approx x'$. (Note that $h_x(x') = x$ even though $x'$ may contain less information than $x$ because $h_x$ is specific to the current input $x$.)

**Definition 1  Additive feature attribution methods** *have an explanation model that is a linear function of binary variables:*

$$g(z') = \phi_0 + \sum_{i=1}^{M} \phi_i z_i', \tag{1}$$

*where $z' \in \{0, 1\}^M$, $M$ is the number of simplified input features, and $\phi_i \in \mathbb{R}$.*

Methods with explanation models matching Definition 1 attribute an effect $\phi_i$ to each feature, and summing the effects of all feature attributions approximates the output $f(x)$ of the original model. Many current methods match Definition 1, several of which are discussed below.

### 2.1    LIME

The *LIME* method interprets individual model predictions based on locally approximating the model around a given prediction [5]. The local linear explanation model that LIME uses adheres to Equation 1 exactly and is thus an additive feature attribution method. LIME refers to simplified inputs $x'$ as "interpretable inputs," and the mapping $x = h_x(x')$ converts a binary vector of interpretable inputs into the original input space. Different types of $h_x$ mappings are used for different input spaces. For bag of words text features, $h_x$ converts a vector of 1's or 0's (present or not) into the original word count if the simplified input is one, or zero if the simplified input is zero. For images, $h_x$ treats the image as a set of super pixels; it then maps 1 to leaving the super pixel as its original value and 0 to replacing the super pixel with an average of neighboring pixels (this is meant to represent being missing).

To find $\phi$, LIME minimizes the following objective function:

$$\xi = \underset{g \in \mathcal{G}}{\arg\min} \ L(f, g, \pi_{x'}) + \Omega(g). \tag{2}$$

Faithfulness of the explanation model $g(z')$ to the original model $f(h_x(z'))$ is enforced through the loss $L$ over a set of samples in the simplified input space weighted by the local kernel $\pi_{x'}$. $\Omega$ penalizes the complexity of $g$. Since in LIME $g$ follows Equation 1 and $L$ is a squared loss, Equation 2 can be solved using penalized linear regression.

## 2.2 DeepLIFT

*DeepLIFT* was recently proposed as a recursive prediction explanation method for deep learning [8, 7]. It attributes to each input $x_i$ a value $C_{\Delta x_i \Delta y}$ that represents the effect of that input being set to a reference value as opposed to its original value. This means that for DeepLIFT, the mapping $x = h_x(x')$ converts binary values into the original inputs, where $1$ indicates that an input takes its original value, and $0$ indicates that it takes the reference value. The reference value, though chosen by the user, represents a typical uninformative background value for the feature.

DeepLIFT uses a "summation-to-delta" property that states:

$$\sum_{i=1}^{n} C_{\Delta x_i \Delta o} = \Delta o, \tag{3}$$

where $o = f(x)$ is the model output, $\Delta o = f(x) - f(r)$, $\Delta x_i = x_i - r_i$, and $r$ is the reference input. If we let $\phi_i = C_{\Delta x_i \Delta o}$ and $\phi_0 = f(r)$, then DeepLIFT's explanation model matches Equation 1 and is thus another additive feature attribution method.

## 2.3 Layer-Wise Relevance Propagation

The *layer-wise relevance propagation* method interprets the predictions of deep networks [1]. As noted by Shrikumar et al., this menthod is equivalent to DeepLIFT with the reference activations of all neurons fixed to zero. Thus, $x = h_x(x')$ converts binary values into the original input space, where $1$ means that an input takes its original value, and $0$ means an input takes the $0$ value. Layer-wise relevance propagation's explanation model, like DeepLIFT's, matches Equation 1.

## 2.4 Classic Shapley Value Estimation

Three previous methods use classic equations from cooperative game theory to compute explanations of model predictions: Shapley regression values [4], Shapley sampling values [9], and Quantitative Input Influence [3].

*Shapley regression values* are feature importances for linear models in the presence of multicollinearity. This method requires retraining the model on all feature subsets $S \subseteq F$, where $F$ is the set of all features. It assigns an importance value to each feature that represents the effect on the model prediction of including that feature. To compute this effect, a model $f_{S \cup \{i\}}$ is trained with that feature present, and another model $f_S$ is trained with the feature withheld. Then, predictions from the two models are compared on the current input $f_{S \cup \{i\}}(x_{S \cup \{i\}}) - f_S(x_S)$, where $x_S$ represents the values of the input features in the set $S$. Since the effect of withholding a feature depends on other features in the model, the preceding differences are computed for all possible subsets $S \subseteq F \setminus \{i\}$. The Shapley values are then computed and used as feature attributions. They are a weighted average of all possible differences:

$$\phi_i = \sum_{S \subseteq F \setminus \{i\}} \frac{|S|!(|F| - |S| - 1)!}{|F|!} \left[ f_{S \cup \{i\}}(x_{S \cup \{i\}}) - f_S(x_S) \right]. \tag{4}$$

For Shapley regression values, $h_x$ maps $1$ or $0$ to the original input space, where $1$ indicates the input is included in the model, and $0$ indicates exclusion from the model. If we let $\phi_0 = f_\varnothing(\varnothing)$, then the Shapley regression values match Equation 1 and are hence an additive feature attribution method.

*Shapley sampling values* are meant to explain any model by: (1) applying sampling approximations to Equation 4, and (2) approximating the effect of removing a variable from the model by integrating over samples from the training dataset. This eliminates the need to retrain the model and allows fewer than $2^{|F|}$ differences to be computed. Since the explanation model form of Shapley sampling values is the same as that for Shapley regression values, it is also an additive feature attribution method.

*Quantitative input influence* is a broader framework that addresses more than feature attributions. However, as part of its method it independently proposes a sampling approximation to Shapley values that is nearly identical to Shapley sampling values. It is thus another additive feature attribution method.

# 3 Simple Properties Uniquely Determine Additive Feature Attributions

A surprising attribute of the class of additive feature attribution methods is the presence of a single unique solution in this class with three desirable properties (described below). While these properties are familiar to the classical Shapley value estimation methods, they were previously unknown for other additive feature attribution methods.

The first desirable property is *local accuracy*. When approximating the original model $f$ for a specific input $x$, local accuracy requires the explanation model to at least match the output of $f$ for the simplified input $x'$ (which corresponds to the original input $x$).

**Property 1 (Local accuracy)**

$$f(x) = g(x') = \phi_0 + \sum_{i=1}^{M} \phi_i x_i' \tag{5}$$

*The explanation model $g(x')$ matches the original model $f(x)$ when $x = h_x(x')$, where $\phi_0 = f(h_x(\mathbf{0}))$ represents the model output with all simplified inputs toggled off (i.e. missing).*

The second property is *missingness*. If the simplified inputs represent feature presence, then missingness requires features missing in the original input to have no impact. All of the methods described in Section 2 obey the missingness property.

**Property 2 (Missingness)**

$$x_i' = 0 \implies \phi_i = 0 \tag{6}$$

*Missingness constrains features where $x_i' = 0$ to have no attributed impact.*

The third property is *consistency*. Consistency states that if a model changes so that some simplified input's contribution increases or stays the same regardless of the other inputs, that input's attribution should not decrease.

**Property 3 (Consistency)** *Let $f_x(z') = f(h_x(z'))$ and $z' \setminus i$ denote setting $z_i' = 0$. For any two models $f$ and $f'$, if*

$$f_x'(z') - f_x'(z' \setminus i) \geq f_x(z') - f_x(z' \setminus i) \tag{7}$$

*for all inputs $z' \in \{0, 1\}^M$, then $\phi_i(f', x) \geq \phi_i(f, x)$.*

**Theorem 1** *Only one possible explanation model $g$ follows Definition 1 and satisfies Properties 1, 2, and 3:*

$$\phi_i(f, x) = \sum_{z' \subseteq x'} \frac{|z'|!(M - |z'| - 1)!}{M!} [f_x(z') - f_x(z' \setminus i)] \tag{8}$$

*where $|z'|$ is the number of non-zero entries in $z'$, and $z' \subseteq x'$ represents all $z'$ vectors where the non-zero entries are a subset of the non-zero entries in $x'$.*

Theorem 1 follows from combined cooperative game theory results, where the values $\phi_i$ are known as Shapley values [6]. Young (1985) demonstrated that Shapley values are the only set of values that satisfy three axioms similar to Property 1, Property 3, and a final property that we show to be redundant in this setting (see Supplementary Material). Property 2 is required to adapt the Shapley proofs to the class of additive feature attribution methods.

Under Properties 1-3, for a given simplified input mapping $h_x$, Theorem 1 shows that there is only one possible additive feature attribution method. This result implies that methods not based on Shapley values violate local accuracy and/or consistency (methods in Section 2 already respect missingness). The following section proposes a unified approach that improves previous methods, preventing them from unintentionally violating Properties 1 and 3.

# 4 SHAP (SHapley Additive exPlanation) Values

We propose SHAP values as a unified measure of feature importance. These are the Shapley values of a conditional expectation function of the original model; thus, they are the solution to Equation

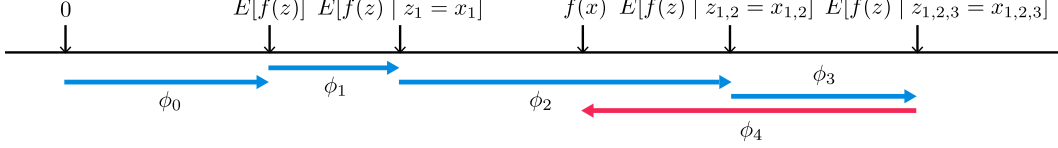

Figure 1: SHAP (SHapley Additive exPlanation) values attribute to each feature the change in the expected model prediction when conditioning on that feature. They explain how to get from the base value $E[f(z)]$ that would be predicted if we did not know any features to the current output $f(x)$. This diagram shows a single ordering. When the model is non-linear or the input features are not independent, however, the order in which features are added to the expectation matters, and the SHAP values arise from averaging the $\phi_i$ values across all possible orderings.

8, where $f_x(z') = f(h_x(z')) = E[f(z) \mid z_S]$, and $S$ is the set of non-zero indexes in $z'$ (Figure 1). Based on Sections 2 and 3, SHAP values provide the unique additive feature importance measure that adheres to Properties 1-3 and uses conditional expectations to define simplified inputs. Implicit in this definition of SHAP values is a simplified input mapping, $h_x(z') = z_S$, where $z_S$ has missing values for features not in the set $S$. Since most models cannot handle arbitrary patterns of missing input values, we approximate $f(z_S)$ with $E[f(z) \mid z_S]$. This definition of SHAP values is designed to closely align with the Shapley regression, Shapley sampling, and quantitative input influence feature attributions, while also allowing for connections with LIME, DeepLIFT, and layer-wise relevance propagation.

The exact computation of SHAP values is challenging. However, by combining insights from current additive feature attribution methods, we can approximate them. We describe two model-agnostic approximation methods, one that is already known (Shapley sampling values) and another that is novel (Kernel SHAP). We also describe four model-type-specific approximation methods, two of which are novel (Max SHAP, Deep SHAP). When using these methods, feature independence and model linearity are two optional assumptions simplifying the computation of the expected values (note that $\bar{S}$ is the set of features not in $S$):

$$
\begin{align}
f(h_x(z')) &= E[f(z) \mid z_S] && \text{SHAP explanation model simplified input mapping} && (9)\\
&= E_{z_{\bar{S}} \mid z_S}[f(z)] && \text{expectation over } z_{\bar{S}} \mid z_S && (10)\\
&\approx E_{z_{\bar{S}}}[f(z)] && \text{assume feature independence (as in [9, 5, 7, 3])} && (11)\\
&\approx f([z_S, E[z_{\bar{S}}]]). && \text{assume model linearity} && (12)
\end{align}
$$

### 4.1 Model-Agnostic Approximations

If we assume feature independence when approximating conditional expectations (Equation 11), as in [9, 5, 7, 3], then SHAP values can be estimated directly using the Shapley sampling values method [9] or equivalently the Quantitative Input Influence method [3]. These methods use a sampling approximation of a permutation version of the classic Shapley value equations (Equation 8). Separate sampling estimates are performed for each feature attribution. While reasonable to compute for a small number of inputs, the Kernel SHAP method described next requires fewer evaluations of the original model to obtain similar approximation accuracy (Section 5).

**Kernel SHAP (Linear LIME + Shapley values)**

Linear LIME uses a linear explanation model to locally approximate $f$, where local is measured in the simplified binary input space. At first glance, the regression formulation of LIME in Equation 2 seems very different from the classical Shapley value formulation of Equation 8. However, since linear LIME is an additive feature attribution method, we know the Shapley values are the only possible solution to Equation 2 that satisfies Properties 1-3 – local accuracy, missingness and consistency. A natural question to pose is whether the solution to Equation 2 recovers these values. The answer depends on the choice of loss function $L$, weighting kernel $\pi_{x'}$ and regularization term $\Omega$. The LIME choices for these parameters are made heuristically; using these choices, Equation 2 does not recover the Shapley values. One consequence is that local accuracy and/or consistency are violated, which in turn leads to unintuitive behavior in certain circumstances (see Section 5).

Below we show how to avoid heuristically choosing the parameters in Equation 2 and how to find the loss function $L$, weighting kernel $\pi_{x'}$, and regularization term $\Omega$ that recover the Shapley values.

**Theorem 2 (Shapley kernel)** *Under Definition 1, the specific forms of $\pi_{x'}$, $L$, and $\Omega$ that make solutions of Equation 2 consistent with Properties 1 through 3 are:*

$$\Omega(g) = 0,$$

$$\pi_{x'}(z') = \frac{(M-1)}{(M \ choose \ |z'|)|z'|(M-|z'|)},$$

$$L(f, g, \pi_{x'}) = \sum_{z' \in Z} \left[f(h_x(z')) - g(z')\right]^2 \pi_{x'}(z'),$$

*where $|z'|$ is the number of non-zero elements in $z'$.*

The proof of Theorem 2 is shown in the Supplementary Material.

It is important to note that $\pi_{x'}(z') = \infty$ when $|z'| \in \{0, M\}$, which enforces $\phi_0 = f_x(\varnothing)$ and $f(x) = \sum_{i=0}^{M} \phi_i$. In practice, these infinite weights can be avoided during optimization by analytically eliminating two variables using these constraints.

Since $g(z')$ in Theorem 2 is assumed to follow a linear form, and $L$ is a squared loss, Equation 2 can still be solved using linear regression. As a consequence, the Shapley values from game theory can be computed using weighted linear regression.[2] Since LIME uses a simplified input mapping that is equivalent to the approximation of the SHAP mapping given in Equation 12, this enables regression-based, model-agnostic estimation of SHAP values. Jointly estimating all SHAP values using regression provides better sample efficiency than the direct use of classical Shapley equations (see Section 5).

The intuitive connection between linear regression and Shapley values is that Equation 8 is a difference of means. Since the mean is also the best least squares point estimate for a set of data points, it is natural to search for a weighting kernel that causes linear least squares regression to recapitulate the Shapley values. This leads to a kernel that distinctly differs from previous heuristically chosen kernels (Figure 2A).

## 4.2 Model-Specific Approximations

While Kernel SHAP improves the sample efficiency of model-agnostic estimations of SHAP values, by restricting our attention to specific model types, we can develop faster model-specific approximation methods.

### Linear SHAP

For linear models, if we assume input feature independence (Equation 11), SHAP values can be approximated directly from the model's weight coefficients.

**Corollary 1 (Linear SHAP)** *Given a linear model $f(x) = \sum_{j=1}^{M} w_j x_j + b$: $\phi_0(f, x) = b$ and*

$$\phi_i(f, x) = w_j(x_j - E[x_j])$$

This follows from Theorem 2 and Equation 11, and it has been previously noted by Štrumbelj and Kononenko [9].

### Low-Order SHAP

Since linear regression using Theorem 2 has complexity $O(2^M + M^3)$, it is efficient for small values of $M$ if we choose an approximation of the conditional expectations (Equation 11 or 12).

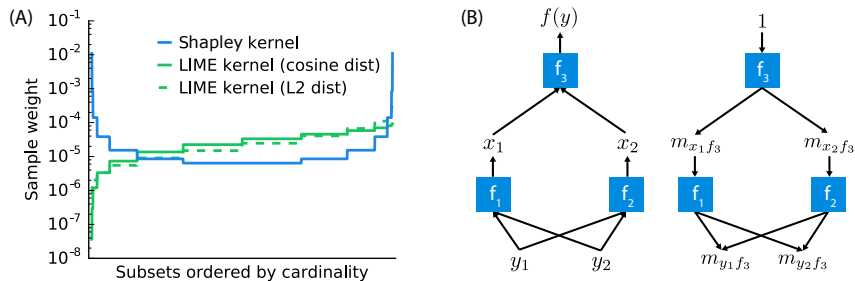

Figure 2: (A) The Shapley kernel weighting is symmetric when all possible $z'$ vectors are ordered by cardinality there are $2^{15}$ vectors in this example. This is distinctly different from previous heuristically chosen kernels. (B) Compositional models such as deep neural networks are comprised of many simple components. Given analytic solutions for the Shapley values of the components, fast approximations for the full model can be made using DeepLIFT's style of back-propagation.

## Max SHAP

Using a permutation formulation of Shapley values, we can calculate the probability that each input will increase the maximum value over every other input. Doing this on a sorted order of input values lets us compute the Shapley values of a max function with $M$ inputs in $O(M^2)$ time instead of $O(M2^M)$. See Supplementary Material for the full algorithm.

## Deep SHAP (DeepLIFT + Shapley values)

While Kernel SHAP can be used on any model, including deep models, it is natural to ask whether there is a way to leverage extra knowledge about the compositional nature of deep networks to improve computational performance. We find an answer to this question through a previously unappreciated connection between Shapley values and DeepLIFT [8]. If we interpret the reference value in Equation 3 as representing $E[x]$ in Equation 12, then DeepLIFT approximates SHAP values assuming that the input features are independent of one another and the deep model is linear. DeepLIFT uses a linear composition rule, which is equivalent to linearizing the non-linear components of a neural network. Its back-propagation rules defining how each component is linearized are intuitive but were heuristically chosen. Since DeepLIFT is an additive feature attribution method that satisfies local accuracy and missingness, we know that Shapley values represent the only attribution values that satisfy consistency. This motivates our adapting DeepLIFT to become a compositional approximation of SHAP values, leading to Deep SHAP.

Deep SHAP combines SHAP values computed for smaller components of the network into SHAP values for the whole network. It does so by recursively passing DeepLIFT's multipliers, now defined in terms of SHAP values, backwards through the network as in Figure 2B:

$$m_{x_j f_3} = \frac{\phi_i(f_3, x)}{x_j - E[x_j]} \tag{13}$$

$$\forall_{j \in \{1,2\}} \quad m_{y_i f_j} = \frac{\phi_i(f_j, y)}{y_i - E[y_i]} \tag{14}$$

$$m_{y_i f_3} = \sum_{j=1}^{2} m_{y_i f_j} m_{x_j f_3} \qquad \text{chain rule} \tag{15}$$

$$\phi_i(f_3, y) \approx m_{y_i f_3}(y_i - E[y_i]) \qquad \text{linear approximation} \tag{16}$$

Since the SHAP values for the simple network components can be efficiently solved analytically if they are linear, max pooling, or an activation function with just one input, this composition rule enables a fast approximation of values for the whole model. Deep SHAP avoids the need to heuristically choose ways to linearize components. Instead, it derives an effective linearization from the SHAP values computed for each component. The $max$ function offers one example where this leads to improved attributions (see Section 5).

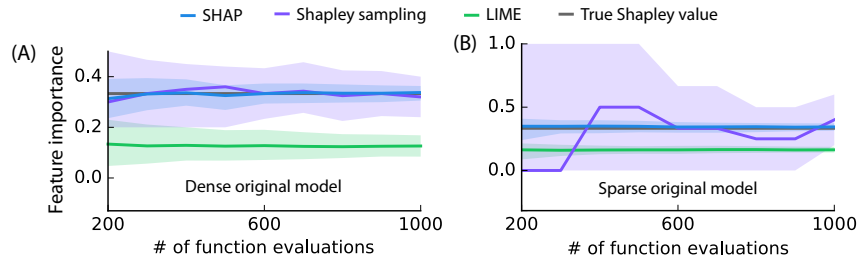

Figure 3: Comparison of three additive feature attribution methods: Kernel SHAP (using a debiased lasso), Shapley sampling values, and LIME (using the open source implementation). Feature importance estimates are shown for one feature in two models as the number of evaluations of the original model function increases. The 10th and 90th percentiles are shown for 200 replicate estimates at each sample size. (A) A decision tree model using all 10 input features is explained for a single input. (B) A decision tree using only 3 of 100 input features is explained for a single input.

# 5 Computational and User Study Experiments

We evaluated the benefits of SHAP values using the Kernel SHAP and Deep SHAP approximation methods. First, we compared the computational efficiency and accuracy of Kernel SHAP vs. LIME and Shapley sampling values. Second, we designed user studies to compare SHAP values with alternative feature importance allocations represented by DeepLIFT and LIME. As might be expected, SHAP values prove more consistent with human intuition than other methods that fail to meet Properties 1-3 (Section 2). Finally, we use MNIST digit image classification to compare SHAP with DeepLIFT and LIME.

## 5.1 Computational Efficiency

Theorem 2 connects Shapley values from game theory with weighted linear regression. Kernal SHAP uses this connection to compute feature importance. This leads to more accurate estimates with fewer evaluations of the original model than previous sampling-based estimates of Equation 8, particularly when regularization is added to the linear model (Figure 3). Comparing Shapley sampling, SHAP, and LIME on both dense and sparse decision tree models illustrates both the improved sample efficiency of Kernel SHAP and that values from LIME can differ significantly from SHAP values that satisfy local accuracy and consistency.

## 5.2 Consistency with Human Intuition

Theorem 1 provides a strong incentive for all additive feature attribution methods to use SHAP values. Both LIME and DeepLIFT, as originally demonstrated, compute different feature importance values. To validate the importance of Theorem 1, we compared explanations from LIME, DeepLIFT, and SHAP with user explanations of simple models (using Amazon Mechanical Turk). Our testing assumes that good model explanations should be consistent with explanations from humans who understand that model.

We compared LIME, DeepLIFT, and SHAP with human explanations for two settings. The first setting used a sickness score that was higher when only one of two symptoms was present (Figure 4A). The second used a max allocation problem to which DeepLIFT can be applied. Participants were told a short story about how three men made money based on the maximum score any of them achieved (Figure 4B). In both cases, participants were asked to assign credit for the output (the sickness score or money won) among the inputs (i.e., symptoms or players). We found a much stronger agreement between human explanations and SHAP than with other methods. SHAP's improved performance for max functions addresses the open problem of max pooling functions in DeepLIFT [7].

## 5.3 Explaining Class Differences

As discussed in Section 4.2, DeepLIFT's compositional approach suggests a compositional approximation of SHAP values (Deep SHAP). These insights, in turn, improve DeepLIFT, and a new version

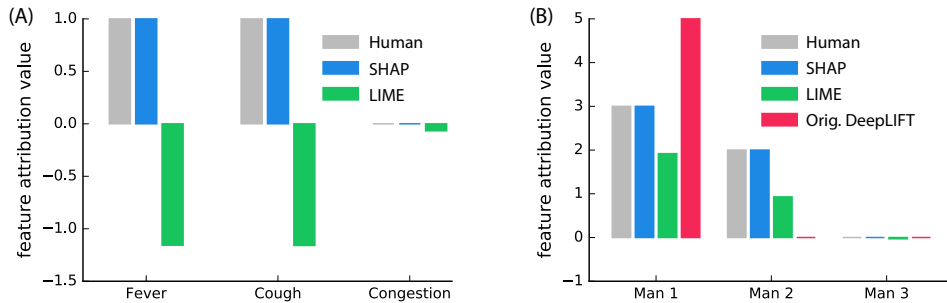

Figure 4: Human feature impact estimates are shown as the most common explanation given among 30 (A) and 52 (B) random individuals, respectively. (A) Feature attributions for a model output value (sickness score) of 2. The model output is 2 when fever and cough are both present, 5 when only one of fever or cough is present, and 0 otherwise. (B) Attributions of profit among three men, given according to the maximum number of questions any man got right. The first man got 5 questions right, the second 4 questions, and the third got none right, so the profit is $5.

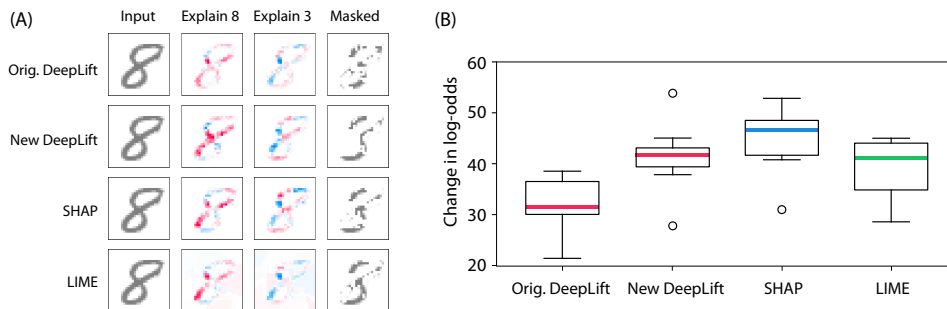

Figure 5: Explaining the output of a convolutional network trained on the MNIST digit dataset. Orig. DeepLIFT has no explicit Shapley approximations, while New DeepLIFT seeks to better approximate Shapley values. (A) Red areas increase the probability of that class, and blue areas decrease the probability. Masked removes pixels in order to go from 8 to 3. (B) The change in log odds when masking over 20 random images supports the use of better estimates of SHAP values.

includes updates to better match Shapley values [7]. Figure 5 extends DeepLIFT's convolutional network example to highlight the increased performance of estimates that are closer to SHAP values. The pre-trained model and Figure 5 example are the same as those used in [7], with inputs normalized between 0 and 1. Two convolution layers and 2 dense layers are followed by a 10-way softmax output layer. Both DeepLIFT versions explain a normalized version of the linear layer, while SHAP (computed using Kernel SHAP) and LIME explain the model's output. SHAP and LIME were both run with 50k samples (Supplementary Figure 1); to improve performance, LIME was modified to use single pixel segmentation over the digit pixels. To match [7], we masked 20% of the pixels chosen to switch the predicted class from 8 to 3 according to the feature attribution given by each method.

# 6  Conclusion

The growing tension between the accuracy and interpretability of model predictions has motivated the development of methods that help users interpret predictions. The SHAP framework identifies the class of additive feature importance methods (which includes six previous methods) and shows there is a unique solution in this class that adheres to desirable properties. The thread of unity that SHAP weaves through the literature is an encouraging sign that common principles about model interpretation can inform the development of future methods.

We presented several different estimation methods for SHAP values, along with proofs and experiments showing that these values are desirable. Promising next steps involve developing faster model-type-specific estimation methods that make fewer assumptions, integrating work on estimating interaction effects from game theory, and defining new explanation model classes.

**Acknowledgements**

This work was supported by a National Science Foundation (NSF) DBI-135589, NSF CAREER DBI-155230, American Cancer Society 127332-RSG-15-097-01-TBG, National Institute of Health (NIH) AG049196, and NSF Graduate Research Fellowship. We would like to thank Marco Ribeiro, Erik Štrumbelj, Avanti Shrikumar, Yair Zick, the Lee Lab, and the NIPS reviewers for feedback that has significantly improved this work.

## Footnotes

[1] https://github.com/slundberg/shap

[2] During the preparation of this manuscript we discovered this parallels an equivalent constrained quadratic minimization formulation of Shapley values proposed in econometrics [2].

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
