[Supplementary Material 1 · exact_max_function.pdf]

# Exact max function Shapley values

August 3, 2017

## 1 Compute the exact Shapley values of the max function in $O(M^2)$ complexity

While the exact Shapley values take $O(2^M)$ time each to compute in general (where $M$ is the number of input features), here we show that for the max function they can be found much more quickly in $O(M^2)$ time to compute all $M$ values. We assume a single reference input, not a whole dataset, so repeating for a set of samples from the dataset would be necessary for computing expectations.

Below is the algorithm (in Julia code) with input vector $x$ and a reference vector $r$. It treats the max function as a decision tree and weights each possible binary outcome with the number of permutations that match it. Note that once a value is fixed that is greater than all other values then the decision tree stops branching. By sorting the inputs by the maximum of their input and reference values we can ensure that one branch of the decision tree will stop branching at each step. Once at a leaf in the tree we compute the effects for all the features encountered so far.

```julia
>>> function shapley_max(x, r)
...
...         # sort so that at each step we know either the input value
...         # or the reference value of the next feature will be the next largest value
...         perm = sortperm(collect(zip(x,r)), by=maximum, rev=true)
...         xsorted = x[perm]
...         rsorted = r[perm]
...
...         M = length(x)
...         path = zeros(M)
...         weight = 1.0
...         num_ones = 0
...         phi = zeros(M)
...         last_val = -Inf
...         weight_scale = 1.0
...
...         for i in 1:M
...             largest_remaining = i == M ? -Inf : max(xsorted[i+1], rsorted[i+1])
...
...             if xsorted[i] >= largest_remaining
...                 path[i] = 1
...                 for j in 1:i
...                     if path[j] == 1
...                         phi[perm[j]] += max(last_val, xsorted[i])*weight*((num_ones+1)/i)/(num_ones+1)
...                     else
...                         phi[perm[j]] -= max(last_val, xsorted[i])*weight*((num_ones+1)/i)/(i-num_ones-1)
...                     end
...                 end
...                 path[i] = 0
...                 weight_scale = (i-num_ones)/i
...             end
...
...             if rsorted[i] >= largest_remaining
```

```
...              path[i] = 0
...              for j in 1:i
...                  if path[j] == 1
...                      phi[perm[j]] += max(last_val, rsorted[i])*weight*((i-num_ones)/i)/num_ones
...                  else
...                      phi[perm[j]] -= max(last_val, rsorted[i])*weight*((i-num_ones)/i)/(i-num_ones)
...                  end
...              end
...              path[i] = 1
...              num_ones += 1
...              weight_scale = num_ones/i
...          end
...
...          if xsorted[i] >= largest_remaining && rsorted[i] >= largest_remaining
...              break
...          end
...
...          last_val = max(min(xsorted[i], rsorted[i]), last_val)
...          weight *= weight_scale
...      end
...
...      phi
... end

shapley_max (generic function with 1 method)
```

## 1.1 Validation using a brute force Shapley method

```
>>> using Iterators
...
... function shapley_weight(M, s)
...     factorial(s)*factorial(M-s-1)/factorial(M)
... end
... function brute_force_shapley(f, x, missing_values, i)
...     phi = 0
...     xtmp = zeros(length(missing_values))
...     for subset in subsets(setdiff(1:length(x), [i]))
...         xtmp[:] = missing_values
...         xtmp[subset] = x[subset]
...         val2 = f(xtmp)
...         xtmp[i] = x[i]
...         val1 = f(xtmp)
...         w = shapley_weight(length(x), length(subset))
...         phi += w*(val1-val2)
...     end
...     phi
... end

brute_force_shapley (generic function with 1 method)
```

```
>>> # test 150 random instances of varying sizes
... for i in 1:15, j in 1:10
...     x = rand(i)
...     r = rand(i)
...     diff = norm(shapley_max(x, r) .- [brute_force_shapley(maximum, x, r, i) for i in 1:length(x)])
...     @assert diff < 1e-8
... end
```



[Supplementary Material 2 · eliminate_symmetry_axiom.pdf]

# The monotonicity axiom implies the symmetry axiom for Shapley values

October 31, 2017

## 1 The monotonicity axiom implies the symmetry axiom for Shapley values

The Shapley values are a well known way in coalitional game theory of assigning credit for an outcome among a set of players. Since any arbitrary mapping between player sets and outcomes is possible, how to assign individual credit can be unclear. The Shapley values are one such way of assigning credit and, importantly, they are the only way that satifies some very basic and desirable properties. Typically these properties are given as four axioms, where the Shapley values are the only credit assignment method that satisfies all four axioms:

1. **Efficiency**

$$f(x) = \sum_{i=0}^{M} \phi_i \tag{1}$$

   This assumption forces the model to correctly capture the original predicted value.
2. **Symmetry**. Let $1_S \in \{0,1\}^M$ be an indicator vector equal to 1 for indexes $i \in S$, and 0 elsewhere, and let $f_x(S) = f(h_x^{-1}(1_S))$. If for all subsets $S$ that do not contain $i$ or $j$

$$f_x(S \cup \{i\}) = f_x(S \cup \{j\}) \tag{2}$$

   then $\phi_i(f, x) = \phi_j(f, x)$. This states that if two features contribute equally to the model then their effects must be the same.
3. **Null effects**. If for all subsets $S$ that do not contain $i$

$$f_x(S \cup \{i\}) = f_x(S) \tag{3}$$

   then $\phi_i(f, x) = 0$. A feature ignored by the model must have an effect of 0.
4. **Linearity**. For any two models $f$ and $f'$

$$\phi_i(f + f', x) = \phi_i(f, x) + \phi_i(f', x). \tag{4}$$

   This states that the effect a feature has on the sum of two functions is the effect is has on one function plus the effect it has on the other.

### 1.1 Young showed in 1985 that linearity and null effects can be eliminated using a monotonicity axiom

**Monotonicity** For any two model functions $f$ and $f'$ if for all subsets $S$ of the simplified input features $Z$ that do not contain $i$

$$f_x(S \cup \{i\}) - f_x(S) \geq f'_x(S \cup \{i\}) - f'_x(S) \tag{5}$$

then

$$\phi_i(f, x) \geq \phi_i(f', x) \tag{6}$$

## 1.2 Here we show how the symmetry axiom is also implied by the monotonicity axiom for models

Assume that $f'$ is the same as $f$ except the inputs $i$ and $j$ are swapped. The means for all subsets $S$ that do not contain $i$ or $j$ that $f'_x(S \cup \{i\}) = f_x(S \cup \{j\})$ and $f'_x(S) = f_x(S)$. If $S$ does contain $j$ then $f'_x(S \setminus \{j\} \cup \{i, j\}) = f_x(S \setminus \{j\} \cup \{j, i\})$, which is garunteed to hold so we can ignore it in the implication below. Starting with the monotonicity axiom

$$\forall_{S \subset Z \setminus \{i\}} \quad f_x(S \cup \{i\}) - f_x(S) \geq f'_x(S \cup \{i\}) - f'_x(S) \implies \phi_i(f, x) \geq \phi_i(f', x) \tag{7}$$

can be transformed into

$$\forall_{S \subset Z \setminus \{i,j\}} \quad f_x(S \cup \{i\}) \geq f_x(S \cup \{j\}) \implies \phi_i(f, x) \geq \phi_i(f', x) \tag{8}$$

by using $f'_x(S \cup \{i\}) = f_x(S \cup \{j\})$, $f'_x(S) = f_x(S)$, and ignoring the terms that include $j$. Swapping $i$ and $j$ and then repreating the process shows that

$$\forall_{S \subset Z \setminus \{i,j\}} \quad f_x(S \cup \{i\}) = f_x(S \cup \{j\}) \implies \phi_i(f, x) = \phi_j(f, x) \tag{9}$$

which is the symmetry axiom. This shows that we only need efficiency and monotonicity to uniquely constrain ourselves to using the Shapley values.



[Supplementary Material 3]

# Shapley Kernel Proof

November 3, 2017

The Shapley kernel is the sample weight given to each binary vector $z' \in \{0,1\}^M$:

$$k(z') = k(M, s) = \frac{M-1}{(M \ choose \ s)s(M-s)}$$

where $s = |z'|$, the number of ones in $z'$.

Let $X$ be the matrix of all possible binary vectors of length $M$ with $2^M$ rows and $M$ columns. We use the Shapley kernel to compute the Shapley values using weighted linear regression:

$$\phi = (X^T W X)^{-1} X^T W y$$

where $W$ is a diagonal matrix with the Shapley kernel weights for each row of $X$, and the $y_i = f_x(S_i)$ values are the function outputs for each row of $X$ (where $S_i$ is the set of ones in $X_{i,*}$). Note that $k(M, 0) = k(M, M) = \infty$, so $W$ is infinity for the all zero row of $X$ and the row of all ones. However, if we set these infinite weights to a large constant, then $X^T W X = \frac{1}{M-1}I + cJ$ for some positive constant $c$ (where $I$ is the identity matrix and $J$ is the matrix of all ones). As $c \to \infty$ the inverted form becomes $(X^T W X)^{-1} = I + \frac{1}{M-1}(I - J)$

The term $X^T W$ is a matrix where all the ones in $X^T$ have been replaced with $k(M, s)$, where $s$ is the number of ones in that column of $X^T$. Multiplying $X^T W$ by $(X^T W X)^{-1}$ creates a matrix of weights to apply to the function outputs in $y$. If we only consider the Shapley value of a single feature $\phi_j$, then we only need to consider a single row of this $2^M \times M$ matrix, which is equivelent to only using the $j$'th row of $(X^T W X)^{-1}$. When we do this we see that the value of the weight for row $i$ is

$$k(M, s_i)\left[\mathbf{1}_{X_{i,j}=1} - \frac{(s_i - \mathbf{1}_{X_{i,j}=1})}{M-1}\right] = \frac{M-1}{(M \ choose \ s_i)s_i(M-s_i)}\mathbf{1}_{X_{i,j}=1} - \frac{(s_i - \mathbf{1}_{X_{i,j}=1})}{(M \ choose \ s_i)s_i(M-s_i)} \tag{1}$$

$$= \frac{(M-1)(M-s_i)!s_i!}{M!s_i(M-s_i)}\mathbf{1}_{X_{i,j}=1} - \frac{(s_i - \mathbf{1}_{X_{i,j}=1})(M-s_i)!s_i!}{M!s_i(M-s_i)} \tag{2}$$

$$= \frac{(M-1)(M-s_i-1)!(s_i-1)!}{M!}\mathbf{1}_{X_{i,j}=1} - \frac{(s_i - \mathbf{1}_{X_{i,j}=1})(M-s_i-1)!(s_i-1)!}{M!} \tag{3}$$

$$= \frac{(M-s_i-1)!(s_i-1)!}{M!}[(M-1)\mathbf{1}_{X_{i,j}=1} - (s_i - \mathbf{1}_{X_{i,j}=1})] \tag{4}$$

where $s_i$ is the number of ones in the $i$'th row of $X$, and $\mathbf{1}_{X_{i,j}=1}$ is one if $X_{i,j} = 1$ and zero otherwise. When $\mathbf{1}_{X_{i,j}=1} = 0$ we get

$$-\frac{(M-s_i-1)!s_i!}{M!}$$

When $\mathbf{1}_{X_{i,j}=1} = 1$ we get

$$\frac{(M-s_i-1)!(s_i-1)!}{M!}[(M-1)-(s_i-1)] = \frac{(M-s_i-2)!(s_i-1)!}{M!}$$

Taking the dot product of these values with $y$ leads to the following equation

$$\phi_j = \sum_{S \subseteq N \backslash j} \frac{(M - s_i - 1)! s_i!}{M!} [f_x(S \cup \{i\}) - f_x(S)]$$

which is a classic form of estimating the Shapley value $\phi_j$ ($N$ is the set of all features).



[Supplementary Material 4]



Figure S1: Convergence of Kernel SHAP estimates for the example image in Figure 5A. Standard deviation is over ten replicate runs of Kernel SHAP for each # of network evaluations considered.