[Reviews · NeurIPS 2017]

Reviewer 1



The authors show that several methods in the literature used for explaining individual model predictions fall into the category of "additive feature attribution" methods. They proposes a new kind of additive feature attribution method based on the concept of Shapely values and call the resulting explanations the SHAP values. The authors also suggest a new kernel called the shapely kernel which can be used to compute SHAP values via linear regression (a method they call kernel SHAP). They discuss how other methods, such as DeepLIFT, can be improved by better approximating the Shapely values. Summary of review: Positives: (1) Novel and sound theoretical framework for approaching the question of model explanations, which has been very lacking in the field (most other methods were developed ad-hoc). (2) Early versions of this paper have already been cited to justify improvements in other methods (specifically New DeepLIFT). (3) Kernel SHAP is a significantly superior way of approximating Shapely values compared to classical Shapely sampling - much lower variance vs. number of model evaluations (Figure 3). Negatives: (1) Algorithm for Max SHAP is incorrect (2) Proof for Kernel SHAP included in supplement is incomplete and hastily written (3) Paper has significant typos: main text says runtime of Max SHAP is O(M^3) but supplement says O(M^2), equation for L under Theorem 2 has a missing close parenthesis for f() (4) Argument that SHAP values are a superior form of model explanation is contentious. Case studies in Figure 4 have been chosen to favor SHAP. Paper doesn't have a discussion of the runtime for kernel SHAP or the recommended number of function evaluations needed for good performance. LIME was not included in the Figure 5 comparison. Detailed comments: The fact that Shapely values can be adapted to the task of model explanations is an excellent insight, and early versions of this paper have already been used by other authors to justify improvements to their methods (specifically New DeepLIFT). Kernel SHAP seems like a very significant contribution because, based on Figure 3, it has far lower variance compared to Shapely sampling when estimating the true Shapely values. It is also appealing because, unlike the popular method LIME, the choice of kernel here is motivated by a sound theoretical foundation. However, there are a few problems with the paper. The algorithm for Max SHAP assumes that when i1 is greater than i2 and i2 is the largest input seen so far, then including i1 contributes (i1 - max(reference_of_i1, i2)) to the output. However, this is only true if none of the inputs which haven't been included yet have a reference value that exceeds max(reference_of_i1, i2). To give a concrete example, imagine there are two inputs, a and b, where a=10 and b=6, and the reference values are ref_a = 9 and ref_b = 0. The reference value of max(a,b) is 9, and the difference-from-reference is 10. The correct SHAP values are 1 for a and 0 for b, because b is so far below the reference of a that it never influences the output. However, the line phi[ind] += max(ref,0)/M will assign a SHAP value of 3 to b (ref = xsorted[i] - r[ind], which is 6-0 for b, and M=2). In the tests, it appears that the authors only checked cases where all inputs had the same reference. While this is often true of maxpooling neurons, it need not be true of maxout neurons. Also, the algorithm seems difficult to implement on a GPU in a way that would not significantly slow down backpropagation, particularly in the case of maxpooling layers. That said, this algorithm is not central to the thesis of the paper and was clearly written in a rush (the main text says the runtime is O(M^3) but the supplement says the runtime is O(M^2)). Additionally, when I looked up the "Shapely kernel proof" in the supplement, the code provided for the computational proof was incomplete and the explanation was terse and contained grammatical errors. After several re-reads, my understanding of the proof is that: (1) f_x(S) represents the output of the function when all inputs except those in the subset S are masked (2) Shapely values are a linear function of the vector of values f_x(S) generated by enumerating all possible subsets S (3) kernel SHAP (which is doing weighted linear regression) also produces an answer that is linear in the vector of all output values f_x(S) (4) therefore, if the linear function in both cases is the same, then kernel SHAP is computing the Shapely values. Based on this understanding, I expected the computational proof (which was performed for functions of up to 10 inputs) to check whether the final coefficients applied to the vector of output values when analytically solving kernel SHAP were the same as the coefficients used in the classic Shapely value computation. In other words, to use the standard formula for weighted linear regression, the final coefficients for kernel SHAP would be the matrix (X^T W X)^{-1} X^T W where X is a binary matrix in which the rows enumerate all possible subsets S and W are the weights computed by the shapely kernel; I expected the ith row of this matrix to match the coefficients for each S in the equation at the top of the page. However, it appears that the code for the computational proof is not comparing any coefficients; rather, for a *specific* model generated by the method single_point_model, it compares the final Shapely values produced by kernel SHAP to the values produced by the classic Shapely value computation. Specifically, the code checks that there is negligible difference between kernel_vals and classic_vals, but classic_vals is returned by the method classic_shapely which accepts as one of its arguments a specific model f and returns the shapely values. The code also calls several methods for which the source has not been provided (namely single_point_model, rawShapely, and kernel_shapely). The statement that SHAP values are a superior form of model explanation is contentious. The authors provide examples in Figure 4 where the SHAP values accord perfectly with human intuition for assigning feature importance, but it is also possible to devise examples where this might not be the case - for example, consider a filter in a convolutional neuron network which has a ReLU activation and a negative bias. This filter produces a positive output when it sees a sufficiently good match to some pattern and an output of zero otherwise. When the neuron produces an output of zero, a human might assign zero importance to all the pixels in the input because the input as a whole did not match the desired pattern. However, SHAP values may assign positive importance to those pixels which resemble the desire pattern and negative importance to the rest - in other words, it might assign nonzero importance to the input pixels even when the input is white noise. In a similar vein, the argument to favor kernel SHAP over LIME would be more compelling if a comparison with LIME were included in Figure 5. The authors state that didn't do the comparison because the standard implementation of LIME uses superpixel segmentation which is not appropriate here. However, I understand that kernel SHAP and LIME are identical except for the choice of weighting kernel, so I am confused why the authors were not able to adapt their implementation of kernel SHAP to replicate LIME. Some discussion of runtime or the recommended number of function evaluations would have been desirable; the primary reason LIME uses superpixel segmentation for images is to reduce computational cost, and a key advantage of DeepLIFT-style backpropagation is computational efficiency. The authors show a connection between DeepLIFT and the SHAP values which has already been used by the authors of DeepLIFT to justify improvements to their method. However, the section on Deep SHAP is problematic. The authors state that "Since the SHAP values for the simple components of the network can be analytically solved efficiently, this composition rule enables a fast approximation of values for the whole model", but they explicitly discuss only 2 analytical solutions: Linear SHAP and Max SHAP. Linear SHAP results in idential backpropagation rules for linear components as original DeepLIFT. As explained earlier, the algorithm for Max SHAP proposed in the supplement appears to be incorrect. The authors mention Low-order SHAP and state that it can be solved efficiently, but Low-order SHAP is not appropriate for many neural networks as individual neurons can have thousands of inputs, and even in the case where a neuron has very few inputs it is unclear if the weighted linear regression can be implemented on a GPU in a way that would not significantly slow down the backpropagation (the primary reason to use DeepLIFT-style backpropagation is computational efficiency). Despite these problems, I feel that the adaptation of Shapely values to the task of model interpretation, as well as the development of kernel SHAP, are substantive enough to accept this paper. REVISED REVIEW: The authors have addressed my concerns and corrected the issues.

Reviewer 2



The paper introduces a united framework for explaining model predictions on a single input. Under the class of additive feature importance, there is a only one solution that satisfies a set of desired properties. Efficient approximation methods to SHAP values are proposed. The authors are tackling a very important problem that draws lots of attention in recent years. The paper is clearly written and it is pleasure to read. The properties proposed for desirable methods are intuitive, though the local accuracy criterion seems too restrictive. It is very novel to use the Sharpley value and give its associated theoretical properties. One sanity check is that when the original model is already interpretable, i.e. the features are 0/1, the method can return the same model. However it seems from Corollary 1 this is not the case since E[x_j] is nonzero. Can the authors provide more explanation? Also, more details about the computation cost(time) of DeepSHAP can be provided to help people have a clear expectation of the method when applying that in practice.

Reviewer 3



Summary: There are many recent efforts in providing interpretability to complex models, such as LIME, DeepLift and Layer-wise relevance propagation. In complex models, change in input features contributes to the output change nonlinearly. Many previous works tried to distribute the feature contributions fairly. In this work, the author pointed that Shapley value -- marginal contribution of each feature -- from game theory is the unique solution that satisfies desired properties. The author provides a unified framework for many of the interpretable methods by introducing the concept of additive feature attribution methods and SHAP as a measurement for feature importance. A SHAP value estimation method is provided. The author supported the method with user studies and comparisons to related works on benchmark datasets. Quality: This paper is technically sound with detailed proofs and supportive experiments. I think there could be more discussion on the shapley value estimation accuracy vs computational efficiency. Clarity: The paper organized the materials well. The author provides enough background and motivations. However, in the implementation of model specific approximations on page 6, section 4.2, it could be more helpful if the author can provide an algorithm flow (for example, how do we change the algorithm in DeepLift to satisfy SHAP values). There are some other minor issues about the clarity: In page 4, equation (10), \bar{S} is not defined, should it be \bar{S} \subseteq S, and the last symbol in the row should be x_{\bar{S}} instead of x? In page 6, line 207, the citation style seems to be inconsistent. Originality: I think this paper nicely pointed out that the proper distribution of the feature contribution is Shapley value from game theory. And the author linked the Shapley value to many existing method to identify the proper way of constructing the methods by proofs. It is interesting that the author explains the enhanced performance in original vs current DeepLift from better approximation in shapley value. I am wondering if the author could also link the shapley value to the gradient-based approaches [1,2], since there are still a lot of interpretations based on these approaches. [1]: Simonyan et al. (2013): "Deep Inside Convolutional Networks: Visualising Image Classification Models and Saliency Maps", http://arxiv.org/abs/1312.6034 [2]: Springenberg et al. (2015): "Striving for Simplicity - The All Convolutional Net", http://arxiv.org/abs/1412.6806 Significance: This work provides a theoretical framework for feature importance methods. The better defined metric SHAP could become a guidance of feature importance evaluation in the research field of interpretable methods.